

Controls of seasonality and altitude on generation of leaf water isotopes
Jinzhao Liu[a, b*], Huawu Wu[c*], Haiwei Zhang[d], Guoqiang Peng[e], Chong Jiang[a], Ying
Zhao[f], Jing Hu[a]
[a] State Key Laboratory of Loess and Quaternary Geology, Institute of Earth Environment,
Chinese Academy of Sciences, Xi'an 710061, China
[b] CAS Center for Excellence in Quaternary Science and Global Change, Xi'an, 710061, China.
[c] Key Laboratory of Watershed Geographic Sciences, Nanjing Institute of Geography and
Limnology, Chinese Academy of Sciences, Nanjing 210008, China
[d] Institute of Global Environmental Change, Xi'an Jiaotong University, Xi'an, 710054, China
[e] School of Geography and Tourism, Shaanxi Normal University, Xi'an 710062, Shaanxi, China
[f] College of resources and environmental engineering, Ludong University, 264025, Yantai,
China
*Corresponding author's email: liujinzhao@ieecas.cn (J. Liu) and wuhuawu416@163.com (H.
Wu)


## Abstract

Stable oxygen and hydrogen isotopes ($\delta^{18}O$ and $\delta^2H$) of leaf water which bridges between hydrological processes and plant-derived organic materials vary spatially and temporally. It is critical to study what controls the $\delta^{18}O$ and $\delta^2H$ values of leaf water for a wide range of applications. Here, we repeatedly sampled soil water, stem water, and leaf water along an elevation transect across seasons on the Chinese Loess Plateau and analyzed the variations in the $\delta^{18}O$ and $\delta^2H$ values from precipitation, soil water, stem water, and leaf water. We found consistency in the $\delta^{18}O$ and $\delta^2H$ values in precipitation, soil water, stem water, and leaf water across seasons, indicating that leaf water can record the isotopic signals of precipitation well. Importantly, leaf water isotope lines were generated by the first-order control of source water (soil water and precipitation) associated with seasonality and altitude, as well as the secondary control of hydroclimate and biochemical factors resulting in weak correlations of the $\delta^{18}O$ and $\delta^2H$ values in leaf water. This study improves our understanding of the generation of leaf water isotopes.

## Short Summary

Why do leaf water isotopes can generate to be an isotopic line in a dual-isotope plot? This isotopic water line is as important as the local meteoric water line (LMWL) in the isotope ecohydrology field. We analyzed the variations of oxygen and hydrogen isotopes in soil water, stem water, and leaf water along an elevation transect across seasons. We found that both seasonality and altitude affecting source water are likely





to result in the generation of an isotopic water line in leaf water.

Keywords: Leaf water, stable isotope, controls, seasonality, altitude

## 1 Introduction
The stable isotope compositions of water ($\delta^{18}O$ and $\delta^2H$) are increasingly used as
powerful tracers to follow the movement of water from its input as precipitation,
movement through the soil, and ultimately to its release as soil evaporation and leaf
transpiration (Mook, 2001; Penna and Meerveld, 2019). Leaf water transpiration plays
a key role in regulating the water balance at scales ranging from catchments to the globe.
Terrestrial plants can enrich heavier isotopes ($^2H$ and $^{18}O$) in leaf water due to
evapotranspiration (Helliker and Ehleinger, 2000; Liu et al., 2015; Cernusak et al.,
2016), which is highly dependent on atmospheric conditions (e.g., temperature and
relative humidity) and biophysiological processes (Farquhar et al., 2007; Kahmen et al.,
2011; Cernusak et al., 2016). Subsequently, the isotope signals of leaf water are
integrated into plant organic materials, such as cellulose (e.g., Barbour, 2007; Lehman
et al., 2017) and leaf wax (Liu et al., 2016, 2021) as powerful proxies used for
paleoclimate reconstruction (Pagani et al., 2006; Schefuβ et al., 2011; Hepp et al.,
2020). Therefore, leaf water $\delta^{18}O$ and $\delta^2H$ values ($\delta^{18}O_{lw}$ and $\delta^2H_{lw}$) are the fundamental
parameters required for an in-depth understanding of these plant organic biomarkers in
paleoclimate contexts.





$\delta^{18}O_{lw}$ and $\delta^2H_{lw}$ values are influenced first by the plant's source water (mainly water
taken up by roots from the soil; Munksgaard et al., 2016; Cernusak et al., 2016). Soil
water for terrestrial plants generally originates from local precipitation, which serves as
a critical component of the water cycle. Precipitation isotopes vary spatially and
temporally, being subject to controls by the temperature, altitude, latitude, distance from
the coast, and amount of precipitation (Bowen, 2010; Bowen and Good, 2015; Cernusak
et al., 2016). Soil water isotopes are determined by a mixture of individual precipitation
events with distinct isotope signals and are also affected by evaporation, both of which
lead to isotopic gradients of soil water with depth (Allison et al., 1983; Liu et al., 2015).
A number of studies have shown that the $\delta^{18}O$ and $\delta^2H$ values of root/xylem water can
be used to characterize the water sources used by plants (Jia et al., 2013; Rothfuss and
Javaux, 2017; Wu et al., 2018; Wang et al., 2019; Amin et al., 2020; Zhao et al., 2020;
Liu et al., 2021a), and these studies rested on an assumption that no isotope
fractionations of $\delta^{18}O$ and $\delta^2H$ values occurred during water uptake by plant roots
(Dawson and Ehleringer, 1991; Ehleringer and Dawson, 1992; Chen et al., 2020),
except in saline or xeric environments (Lin and Sternberg, 1993; Ellsworth and
Williams, 2007). Some recent studies showed, however, that the occurrence of isotopic
fractionation during root water uptake was likely more common, especially regarding
the $\delta^2H$ values (Zhao et al., 2016; Wang et al., 2017; Barbeta et al., 2019; Poca et al.,
2019; Liu et al., 2021a). Therefore, studying the isotopic variations in the water
continuum from precipitation, soil water, stem water, to leaf water will help provide
substantial insight into understanding the spatiotemporal variations in leaf water



isotopes.

In addition to being influenced by plant source water, $\delta^{18}O_{lw}$ and $\delta^2H_{lw}$ values are
influenced by the evaporative process of transpiration and the isotopic composition of
the vapour in the atmosphere surrounding the leaf, the influences of which can be
predicted using the Craig-Gordon model (Craig and Gordon, 1965), which has been
modified for leaves under steady-state conditions (Dongmann et al., 1974; Farquhar et
al.,1989; Farquhar and Cernusak, 2005):
$$\Delta_e = \varepsilon^+ + \varepsilon_k + (\Delta_v - \varepsilon_k)\frac{e_a}{e_i}$$
where $\Delta_e$ is the enrichment of evaporative site water above source water, $\varepsilon^+$ is the
equilibrium fractionation between liquid water and vapour, $\varepsilon_k$ is the kinetic
fractionation during the diffusion of vapour through the stomata and the boundary layer,
$\Delta_v$ is the isotopic enrichment of vapour compared to source water, and $(\frac{e_a}{e_i})$ is the ratio
of the water vapour pressure fraction in the air relative to that in the intercellular spaces,
which is equal to the relative humidity in the air. However, the model fails to explain
the intra-leaf heterogeneity of $\delta^{18}O_{lw}$ and $\delta^2H_{lw}$ (Cernusak et al., 2016; Liu et al., 2021b),
which is currently explained by a two-pool model (Leaney et al., 1985; Song et al.,
2015a) and/or an advection-diffusion model, as the *Péclet* effect (Farquhar and Lloyd,
1993; Farquhar and Gan, 2003). Subsequently, more complicated models under non-
steady-state conditions have been developed (Cuntz et al., 2007; Ogée et al., 2007).
These models emphasize on a mechanistic understanding of leaf water isotopic
fractionation, but under natural conditions, the relevant parameters cannot be strictly





constrained or precisely monitored which hinders the uses of these models (Plavcová
et al., 2018). Thus, the relationships between $\delta^{18}O_{lw}$ and/or $\delta^2H_{lw}$ and geo-climate
factors critically need to be resolved.

In this study, we measured the $\delta^{18}O_{lw}$ and $\delta^2H_{lw}$ values of precipitation, soil water, stem
water, and leaf water along an elevation transect across different seasons (i.e., spring,
summer, autumn). The objectives of our study were to better understand the seasonal
patterns of the leaf water $\delta^{18}O_{lw}$ and $\delta^2H_{lw}$ values, and the controls of altitude and
seasonality on the $\delta^{18}O_{lw}$ and $\delta^2H_{lw}$ values of leaf water generation. The results can help
to qualitatively and quantitatively evaluate the leaf water-based transpiration flux in
ecohydrological processes and the accuracy of plant organic biomarkers in natural
archives since they provide an insight into potential controls on leaf water isotopic
generation.

## 2 Materials and Methods
### 2.1 Study area
The Qinling Mountains form the dividing line between northern and southern China
and mark the boundary between the watersheds of the Yellow River and Yangtze River
valleys. Mt. Taibai (Fig. 1; 33. 96∘ N, 107.77∘ E; 3767 m asl) is the peak of the Qinling
Mountains, with a warm temperate ecosystem characterized by a rich and colourful
flora and fauna. The mean annual temperature at the bottom of Mt. Taibai is 12.9°C,
and the mean annual precipitation is 609.5 mm (Zhang and Liu, 2010). The climate,


soil, and vegetation vary significantly along the slope transect, exhibiting vertical geo-
ecological zonation (Fig. 1), which includes a variety of climate zones: warm temperate
(< 1300 m), temperate (1300 ~ 2600 m), cool temperate (2600 ~ 3350 m), and alpine (>
3350 m). The soil background covers yellow loess soil at low elevations, spectacular
rocky outcrops at middle elevations, and glacial remnants at high elevations; and the
vegetation consists mainly of coniferous and broadleaf forests, and alpine and subalpine
vegetation along the transect.
2.2 Sampling strategy
Plant and soil samples were performed in May, July and September 2020, and the
samples were collected from ten plots (3 × 3 m) along the northern slope of Mt. Taibai
extending from 608 m to 3533 m asl (Fig. 1). One or two plant species were
simultaneously collected; plant species were chose if they had a high abundance in the
community and/or were widely distributed for each plot. For plants, three stem and leaf
samples were collected for each species. Intact leaves with minimal damage were
collected from fully sunlit canopy branches considering the likely isotopic gradients
within a leaf (Liu et al., 2016). Suberized twigs were cut into 3-4 cm segments as a
sample, and these small plant segments were immediately placed into capped glass vials.
For soils, three surface soil samples (less than 10 cm) around the sampling plants were
taken using a small metal scoop. All sampled plots were located on slopes far away
from rivers and surface water bodies, which guaranteed that the soil water in each plot
was derived exclusively from precipitation. Although the surface soil layers were only
collected in this study, these samples provided a comparative reference for soil water





that originated from surface water instead of deep water, which is supported by a prior
study conducted at the same elevation transect (Zhang and Liu, 2010). The soil samples
were tightly sealed in a polyethylene zipper bag on site. All plant and soil samples were
frozen in a cooler (~ 4 °C) in the field and immediately transported to the laboratory.
The altitude of each plot was determined using a handheld GPS unit with an error of $\pm$
5 m.
2.3 Isotopic analyses
Water in plant and soil samples was extracted using an automatic cryogenic vacuum
extraction system (LI-2100 Pro, LICA United Technology Limited, Beijing, China).
The auto-extraction process was set for 3 hours, and the extraction rate of water from
samples was more than 98%. The isotopic composition of soil water was measured
using a Picarro L2130-I isotope water analyzer (Sunnyvale, CA, USA) at the State Key
Laboratory of Loess and Quaternary Geology, Institute of Earth Environment, Chinese
Academy of Science. The analytical accuracies were $\pm$ 0.1‰ for $\delta^{18}O$ and $\pm$ 1‰ for
$\delta^2H$. The isotopic measurements of root and leaf waters were conducted using an
isotope ratio mass spectrometer coupled with a high-temperature conversion elemental
analyzer (HT2000 EA-IRMS, Delta V Advantage; Thermo Fisher Scientific, Inc. USA)
in Huake Precision Stable Isotope Laboratory at the campus of Tsinghua University
Shenzhen International Graduation School. The measurement precisions were $\pm$ 0.2‰
and $\pm$ 1‰ for $\delta^{18}O$ and $\delta^2H$, respectively. The isotopic composition of $\delta^{18}O$ and $\delta^2H$ is
expressed as an isotope ratio:
$\delta_{sample}(‰) = \left(\frac{R_{sample} - R_{standard}}{R_{standard}}\right) \times 1000$      (1)


where $\delta_{sample}$ represents $\delta^{18}O$ or $\delta^2H$, and $R_{sample}$ and $R_{standard}$ indicate the ratio
of $^{18}O/^{16}O$ or $^2H/^1H$ of the sample and standard, respectively. The $\delta^{18}O$ and $\delta^2H$ values
are reported relative to the Vienna mean standard ocean water (VSMOW). The $\delta^{18}O$
and $\delta^2H$ values of precipitation were determined by the Online Isotope in Precipitation
Calculator (Bowen and Revenaugh, 2003).
2.4 Data analysis
Statistical analyses (e.g., mean, max., min., and s.d.) for isotopes of precipitation, soil,
stem and leaf waters were performed to show the range and distribution of $\delta^{18}O$ and
$\delta^2H$. Pearson correlation was conducted to describe the various correlations between
$\delta^{18}O$ and $\delta^2H$ among the different water types (e.g., precipitation, soil water, stem water,
and leaf water) because the isotopic data were normally distributed according to the
Kolmogorov-Smirnov (K-S) test. One-way ANOVA combined with a post hoc Tukey's
least significant difference (LSD) test was performed to identify the significant
differences in isotopic compositions of precipitation, soil, stem, and leaf waters across
months. Comparisons of the relationships of $\delta^{18}O$ and $\delta^2H$ for soil water and leaf water
were performed by using analysis of covariance (ANCOVA) to compare slopes across
months. The significance level for all statistical tests was set to the 95% confidence
interval. Moreover, the Hybrid Single-Particle Lagrangian Integrated Trajectory
(HYSPLIT) model (Draxler and Rolph, 2003) was used to perform air mass back-
trajectory calculations for a central site (34.13°N, 107.83°E, 2270 m asl) of the study
area. Trajectories were initiated four times daily (at 00:00, 06:00, 12:00, and 18:00 UTC)
and their air parcel was released at 2300 m asl for May, July and September 2020 and



moved backward by winds for 120 h (5 days).

## 3 Results

### 3.1 Seasonal variations

The $\delta^{18}O$ and $\delta^2H$ values in precipitation, soil water, stem water, and leaf water varied

significantly among them across months (Fig. 2). The $\delta^{18}O$ values of precipitation, soil

water, stem water, and leaf water were -7.7 ± 2.0‰, -3.8 ± 3.4‰, 1.9 ± 4.2‰, and 4.7

± 3.0‰ in May, -9.1 ± 1.4‰, -11.5 ± 1.3‰, 3.4 ± 2.6‰, and 1.4 ± 4.1‰ in July, and -

9.5 ± 1.5‰, -11.6 ± 1.3‰, 7.5 ± 6.7‰, and 9.7 ± 5.6‰ in September, respectively.

Likewise, the $\delta^2H$ values of precipitation, soil water, stem water, and leaf water were -

50.7 ± 13.9‰, -47.1 ± 10.5‰, -31.6 ± 20.3‰, and -18.2 ± 14.5‰ in May, -63.0 ± 9.7‰,

-92.1 ± 9.9‰, -69.9 ± 15.2‰, and -54.4 ± 12.8‰ in July, and -66.4 ± 10.6‰, -87.7 ±

11.9‰, -84.5 ± 25.2‰, and -97.0 ± 28.0‰ in September, respectively. Both the $\delta^{18}O$

and $\delta^2H$ values for all four water types (i.e., precipitation, soil water, stem water, and

leaf water) were significantly different ($p < 0.05$), exhibiting relatively heavier values

in May, intermediate values in July, and lower values in September, except for the $\delta^{18}O$

values in soil water (Fig. 2).

### 3.2 Correlations of $\delta^{18}O$ and $\delta^2H$ values

Significant correlations of the $\delta^{18}O$ and $\delta^2H$ values in different water types were

observed across months (Fig. 3). The local meteoric water lines (LMWLs) were

obtained from the $\delta^{18}O$ and $\delta^2H$ values of precipitation, in which of the slopes and

intercepts varied slightly across months (7.04, 6.79 and 6.85 for slopes and 3.26, -1.12





and -1.42 for intercepts in May, July and September, respectively). Similarly, the
regression lines of the $\delta^{18}O$ and $\delta^2H$ values from soil water, stem water, and leaf water
were observed (Fig. 3), suggesting that leaf water isotopes could well inherit the
isotopic signals of source waters that originated from stem water, soil water, and
ultimately precipitation. However, the slopes and coefficients of determination ($R^2$) of
the $\delta^{18}O$ and $\delta^2H$ values showed consistent decreasing trends from precipitation, soil
water, stem water and leaf water in all three months, except for soil water in May (Fig.
3). The ANCOVA tests showed no significant differences for the regression lines for
precipitation (df = 0.47, $F$ = 2.49, $p$ = 0.11 > 0.05), stem water (df = 53.2, $F$ = 0.42, $p$
= 0.66 > 0.05), and leaf water (df = 437.3, $F$ = 2.78, $p$ = 0.08 > 0.05) across months,
but a significant difference for soil water across months (df = 308.8, $F$ = 10.9, $p$ < 0.05).
The difference in soil water regression lines across months was probably due to the
mixture of various precipitation events and evaporation in the upper soil layers (Yang
and Fu, 2017).
3.3 Altitude effects
Both the $\delta^{18}O$ and $\delta^2H$ values in precipitation and soil water decreased significantly
with an increase of altitude (Fig. 4; all for $R^2$ > 0.37, $p$ < 0.05). Stem water and leaf
water showed decreasing trends for $\delta^2H$ in July ($R^2$ = 0.82 and 0.43) and for $\delta^{18}O$ ($R^2$
= 0.61 and 0.44) and $\delta^2H$ values ($R^2$ = 0.84 and 0.90) in September; in contrast, there
were nonsignificant correlations between isotopes in stem water and leaf water and
altitude for $\delta^{18}O$ values in July and for both $\delta^{18}O$ and $\delta^2H$ values in September (Fig. 4).



## 4 Discussion

### 4.1 Consistent variations among water types

We found a seasonal consistency in the $\delta^{18}O$ and $\delta^2H$ values from precipitation to soil water, stem water, and ultimately leaf water (Fig. 2). This finding of temporal consistency among water types (i.e., precipitation, soil water, stem water, leaf water) has been observed in a number of studies (Phillips and Ehleringer, 1995; Cernusak et al., 2005; Sprenger et al., 2016; Berry et al., 2017; Liu et al., 2021a). The isotopic inheritance from precipitation to leaf water indicated that seasonal variations in the precipitation $\delta^{18}O$ and $\delta^2H$ values could exert the first order of control on the temporal patterns of leaf water. The spatiotemporal variability of the precipitation $\delta^{18}O$ and $\delta^2H$ values could be explained by a combination of effects such as temperature, altitude, latitude, continent, and amount, which are associated with orographic conditions, sub-cloud evaporation, moisture recycling, and differences in the vapor source at the regional and continental scales (Dansgaard, 1964; McGuire and McDonnell, 2007; Li et al., 2016; Penna and Meerveld, 2019; Wu et al.,2019). Our recent study, conducted approximately 200 km away from the observed transect on the Chinese Loess Plateau, demonstrated that the temperature effect (i.e., altitude effect), but not the precipitation amount effect, was the dominant control on the precipitation $\delta^{18}O$ and $\delta^2H$ values (Liu et al., 2021a). The underlying mechanism of the temperature effect in monsoon regions is very complicated because three typical processes coexist: 1) the evaporation condition over the vapour source area affects the initial isotopic ratio of atmospheric moisture; 2) the transportation process of the water vapour affects the extent of rainout




of the air mass during the course of transportation; and 3) the extent of condensation of
the vapour is influenced by the condensation temperature (Pang et al., 2006; Li et al.,

267    2019).


The $\delta^{18}O$ and $\delta^{2}H$ values from soil water, stem water and leaf water were isotopically
heavier in May, intermediate in July, and lowest in September, responding well to the
decreasing trends of precipitation $\delta^{18}O$ and $\delta^{2}H$ values (Fig. 2). The monthly variations
in precipitation $\delta^{18}O$ and $\delta^{2}H$ values from the Global Network for Isotopes in
Precipitation (GNIP, http://www.iaea.org/) at Xi'an station (1985-1992 AD), ca. 100
km away from our study transect, were $^{18}O$- and $^{2}H$-enriched in May relative to July
and September (Fig. 5a, b). The cluster mean of moisture transport routes using
HYSPLIT (Draxler and Rolph, 2003) and climatological 850 hPa wind vectors showed
that the main moisture was from western China and central Asia in May, from the
China-India Peninsula and the Bay of Bangle, and from the local moisture recycling
and convection (Fig. 5c, d, e). The seasonal variation of precipitation $\delta^{18}O$ and $\delta^{2}H$
values is consistently related to the onset, advancement and retreat of the Asian summer
monsoon and associated large-scale monsoon circulation change (e.g., Cheng et al.,
2009; Zhang et al., 2020, 2021). As the summer monsoon starts in mid-May, the rainfall
season starts in southern China, however, the study area is mainly controlled by the
moisture    from    westerlies    (Chiang    et    al.,    2015)    with    relatively    higher
vapour/precipitation $\delta^{18}O$ and $\delta^{2}H$ values (Fig. 5c, a, b). In July, the summer monsoon
reaches its strongest phase, the rainfall belt shifts to central and northern China, and the





southerly wind brings plenty of moisture from the China-India Peninsula and the Bay
of Bangle with lower vapour/precipitation $\delta^{18}O$ and $\delta^2H$ values (Fig. 5d, a, b). When
the summer monsoon withdraws in September, the study area is mainly controlled by
moisture from local moisture recycling and convection (Fig. 5e). Soil water stores June-
August monsoon rainfall with lower $\delta^{18}O$ and $\delta^2H$ values, resulting in further lower
precipitation $\delta^{18}O$ and $\delta^2H$ values in September than those in July (Fig. 5a, b), and thus
resulting in significantly lower $\delta^{18}O_{lw}$ and $\delta^2H_{lw}$ values (Fig. 6).

4.2 Generation of leaf water isotope line
The LMWL, generated by the precipitation $\delta^{18}O$ and $\delta^2H$ values at the observed
locations (Fig. 2), is an important reference line for ecohydrological process and acts
as a benchmark for comparison among different water types. The LMWLs in May
(spring), July (summer) and September (autumn) were slightly smaller than the global
meteoric water line (GMWL: $\delta^2H = 8.17 \times \delta^{18}O + 10.35$; Rozanski et al., 2013),
suggesting different water vapour sources in the local circulation system and strong
evaporative fractionation under arid conditions. Across months, July precipitation
tended to have an isotopically lower slope value (6.79) than that of both May (7.04) and
September precipitation (6.85), but the difference across months was not significant ($p$
$= 0.11 > 0.05$). The slopes of the LMWLs from different months indicated that the water
vapour of precipitation across seasons came from the same source but suffered from
different intensities of evaporation due to temperature (Wu et al., 2019; Li et al., 2019).
Likewise, the regression lines of the $\delta^{18}O$ and $\delta^2H$ values in soil water, stem water, and





leaf water were observed across months (Fig. 3). The slopes in other water types (i.e,
soil water, stem water, and leaf water) were relatively lower than the LMWLs, in which
the slopes for soil water and stem water were intermediates; however, they were lowest
for leaf water across seasons, except for soil water in May (Fig. 3). These observations
were supported by a variety of studies (Brooks et al., 2010; Evaristo et al., 2015;
Sprenger et al., 2016, 2017; Wang et al., 2017; Benettin et al., 2018; Barbeta et al., 2019;
Penna and Meerveld, 2019; Liu et al., 2021a) due to the occurrence of secondary
evaporation in other water types. Moreover, the $R^2$ values of dual-isotope space ($\delta^{18}$O
and $\delta^2$H) decrease significantly from precipitation, soil water, stem water and leaf water
in all seasons (Fig. 3), suggesting that besides physically evaporative fractionation,
other factors likely affect the $\delta^{18}$O and $\delta^2$H values in leaf water. Although the
abovementioned Craig-Gordon model has been used to explain the variation in $\delta^{18}$O
and $\delta^2$H values in leaf water, the factors that control the leaf water $\delta^{18}$O and $\delta^2$H values
under non-steady-state conditions and the *Péclet* effect remain to be further studied
(Song et al., 2015b; Cernusak et al., 2016; Barbour et al., 2017).
In a dual-isotope space in leaf water, a significantly distributed pattern across months
was observed: isotopically depleted in September, intermediate in July, and enriched in
May (Fig. 6). When focusing on each month, we found relatively higher isotopic values
occurring at low elevations but lower isotopic values at high elevations despite no or
weak correlations between altitude and $\delta^{18}$O/$\delta^2$H values (Fig. 4). Combining these two
effects (i.e., seasonality and altitude), the $\delta^{18}$O and $\delta^2$H values in leaf water yield a
remarkable isotopic line in the dual-isotope plot (Fig. 3), which typically lies at the right



of the LMWLs. This result is supported by a recent study that conducted consecutive
measurements of $\delta^{18}O$ and $\delta^{2}H$ values in xylem/leaf water in Switzerland and indicated
that leaf water provides great potential to determine source water of plants (Benettin et
al., 2021). A schematic of effects of seasonality and altitude on leaf water $\delta^{18}O$ and $\delta^{2}H$
values is shown in Fig. 7, which involves many of hydroclimatic and biochemical
factors that control the leaf water $\delta^{18}O$ and $\delta^{2}H$ values. Significant isotopic fractionation
occurred mainly at two key locations across vertical soil profiles and leaf architectures
from precipitation to leaf water, but both seasonality and altitude, in essence, affected
the precipitation $\delta^{18}O$ and $\delta^{2}H$ values (Fig. 7). An isotopic gradient across the vertical
soil profile appeared because of evaporation at the surface soil layers (Ehleringer et al.,
1992; Goldsmith et al., 2012; Evaristo et al., 2015), which led to a linear enrichment
trajectory in the soil water dual-isotope plot (Goldsmith et al., 2012; Jia et al., 2013;
Rothfuss and Javaux, 2017; Wu et al., 2018; Wang et al., 2019; Amin et al., 2020; Zhao
et al., 2020; Liu et al., 2021a). The soil water isotope line provides a water source for
leaf water isotope line generation. However, biochemical factors also exert an effect on
leaf water $\delta^{18}O$ and $\delta^{2}H$ values, as supported by different $\delta^{2}H$ enrichments in leaf water
between dicots and monocots, associated with leaf veinal structures (Liu et al., 2021b).
This result is consistent with the weaker correlations of $\delta^{18}O$ and $\delta^{2}H$ values in leaf
water than in soil water (Fig. 3). Collectively, the leaf water isotope line is generated
by the first-order control of spatiotemporal variation in the precipitation $\delta^{18}O$ and $\delta^{2}H$
values (associated with seasonality and altitude) and secondarily affected by
biochemical factors within a leaf.






4.3 Insights and implications
The ecohydrological cycle over continents primarily involves the input from
precipitation and the output to the atmosphere through evapotranspiration. Among them,
leaf water transpiration is a key component of water cycle in terrestrial ecosystems.
Stable isotope technique of leaf water has been used to estimate transpiration through
leaf surface, contributing up to 50 to 90% of ecosystem evapotranspiration (Jasechko
et al., 2013; Schlesinger and Jasechko, 2014). Moreover, the $\delta^{18}O$ values of leaf water
partly influences the oxygen isotope values of atmospheric $CO_2$ (Farquhar et al., 1993),
which can be helpful to constrain global carbon cycle. All of these various applications
rest on a firm understanding of the mechanisms that control leaf water $\delta^{18}O$ and $\delta^{2}H$
values (Cernusak et al., 2016).

However, there were great variabilities in leaf water $\delta^{18}O$ and $\delta^{2}H$ values over diurnal
and seasonal cycle. For example, leaf water $\delta^{18}O$ and $\delta^{2}H$ values generally showed a
maximum in the early afternoon and a minimum in the early morning (Cernusak et al.,
2016). Our results showed a seasonal variation in leaf water $\delta^{18}O$ and $\delta^{2}H$ values, which
followed the isotopic patterns in other waters such as stem water, soil water and
precipitation. The seasonal variability in leaf water $\delta^{18}O$ and $\delta^{2}H$ values has also been
observed in tropical monsoon condition (Hartsough et al., 2008). The diurnal and
seasonal variations in leaf water $\delta^{18}O$ and $\delta^{2}H$ values indicate the leaf water isotopic
line would vary with time. Moreover, as we all known, the LMWL varys significantly

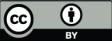



over space, with the slopes ranging between 5 and 6.5 in middle latitudes and between
2 and 5 in arid climates (Gibson et al., 2008; Sprenger et al., 2016). Collectively, the
leaf water isotope line will vary temporally and spatially. Thus, it needs to be widely
explored for the spatio-temporal variations of leaf water isotope line in the future
studies.

## 5 Conclusion
Along the elevation transect, precipitation, soil water, stem water, and leaf water were
repeatedly sampled to analyze for $\delta^{18}O$ and $\delta^2H$ values associated with season and
altitude. There was a seasonal consistency of $\delta^{18}O$ and $\delta^2H$ values from precipitation,
soil water, stem water, and ultimate leaf water, suggesting that leaf water recorded well
the isotopic signals of precipitation, which was primarily affected by the water vapour
source in our studied transect. Moreover, both $\delta^{18}O$ and $\delta^2H$ values of precipitation and
soil water were significantly correlated with altitude, but no or weak correlations
occurred between $\delta^{18}O$ and $\delta^2H$ values of stem/leaf water and altitude, which indicated
that besides source water (i.e., precipitation, soil water), biochemical factors likely
exerted a secondary control on leaf water $\delta^{18}O$ and $\delta^2H$ values. Therefore, leaf water
isotopes were controlled by combined effects of source water and leaf water
transpiration surrounding the leaf surface, in which seasonality and altitude acted as a
trigger to form a leaf water isotopic line.

**Data availability statement**



Data related to this article can be found in Electric Annex and Mendeley Data
(https://data.mendeley.com/drafts/t44wybgpr3).

**Author contribution**
J.L. conceived the idea of research. J.L. and H.W. performed the data analysis and wrote
the manuscript. H.W. and H.Z. edited the paper. C.J. and J.H. performed the lab work.
All authors contributed to discuss the results.

**Competing interests**
The authors declare that they have no known competing financial interests or
personal relationships that could have appeared to influence the work reported in this
paper.

**Acknowledgement**
We would like to thank X. Cao and M. Xing for their help with laboratory assistance
and Y. Cheng for helps in the field. We thank Prof. J. J. McDonnell for pre-discussing
the framework of the paper. This work was supported by the Chinese Academy of
Sciences (XAB2019B02; ZDBS-LY-DQC033) and National Natural Science
Foundation of China (42073017), and by the open fund from State Key Laboratory of
Loess and Quaternary Geology (SKLLQG1926).

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

Contributions of local terrestrial evaporation and transpiration to precipitation using
$\delta^{18}O$ and D-excess as a proxy in Shiyang inland river basin in China. Global and
Planetary Change 146, 140-151.
Li Z, Li Z, Yu H, Song L, Ma J, 2019. Environmental significance and zonal
characteristics of stable isotope of atmospheric precipitation in arid Central Asia.
Atmospheric Research 227, 24-40.
Lin, G.H., Sternberg, L.S.L., 1993. Hydrogen isotopic fractionation by plant roots
during water uptake in coastal wetland plants. Stable Isotopic and Plant Carbon/Water
Relations. Academic Press, New York, pp. 497–510.
Liu, J., Liu, W., An, Z., 2015. Insight into the reasons of leaf wax $\delta D_{\text{n-alkane}}$ values
between grasses and woods. Sci. Bull. 60, 549–555.
Liu, J., Liu, W., An, Z., Yang, H., 2016. Different hydrogen isotope fractionations
during lipid formation in higher plants: Implications for paleohydrology. Sci. Report 6,

544    19711.

Liu J, Wu H, Cheng Y, Jin Z, Hu J, 2021a. Stable isotope analysis of soil and plant water
in a pair of natural grassland and understory of planted forestland on the Chinese Loess
Plateau. Agricultural Water Management 249, 106800.
Liu J, An Z, Lin G, 2021b. Intra-leaf heterogeneities of hydrogen isotope compositions
in leaf water and leaf wax of monocots and dicots. Science of the Total Environment

550    770, 145258.



McGuire, K., and J. McDonnell (2007), Stable isotope tracers in watershed hydrology,
in Stable Isotopes in Ecology and Environmental Science, Ecological Methods and
Concepts Series, pp. 334–374.
Munksgaard NC, Cheesman AW, English NB, Zwart C, Kahmen A, Cernusak LA, 2016.
Identifying drivers of leaf water and cellulose stable isotope enrichment in Eucalyptus
in northern Australia. Oecologia 183, 31-43.
Pang, H., He, Y., Lu, A., Zhao, J., Ning, B., Yuan, L., Song, B., 2006. Synoptic-scale
variation of $\delta^{18}O$ in summer monsoon rainfall at Lijiang, China. Chin. Sci. Bull. 51,
2897–2904.
Pagani M, Pedentchouk N, Huber M, Sluijs A, Schouten S, Brinkhuis H, Damsté J S S.
Dichens GR, 2006. Arctic hydrology during global warming at the Palaeocene/Eocene
thermal maximum. Nature 442, 671–675.
Penna D, van Meerveld HJ, 2019. Spatial variability in the isotopic composition of
water in small catchments and its effect on hydrograph separation. WIREs Water, e1367.
Phillips, S.L., Ehleringer, J.R., 1995. Limited uptake of summer precipitation by big
tooth maple (*Acer grandidentatum* Nutt) and Gambels oak (*Quercus gambelii* Nutt).
Trees 9, 214–219.
Plavcová L, Hronková M, Šimková M, Květoň J, Vráblová M, Kubásek J, Šantrůček J,
2018. Seasonal variation of $\delta^{18}O$ and $\delta^2H$ in leaf water of *Fagus sylvatica* L. and related
water compartments. Journal of Plant Physiology 227, 56-65.
Poca M, Coomans O, Urcelay C, Zeballos SR, Bodé S, Boecks P, 2019. Isotope
fractionation during root water uptake by *Acacia caven* is enhanced by arbuscular



mycorrhizas. Plant and Soil, 3.
Rienecker, M.M., Suarez, M.J., Gelaro, R., Todling, R., Bacmeister, J., Liu, E.,
Bosilovich, M.G., Schubert, S.D., Takacs, L., Kim, G.-K., 2011. MERRA: NASA's
modern-era retrospective analysis for research and applications. Journal of Climate 24,

577     3624-3648.

Rothfuss, Y., Javaux, M., 2017. Reviews and syntheses: isotopic approaches to quantify
root water uptake: a review and comparison of methods. Biogeosciences 14, 2199–2224.
Rozanski, K., Araguas-Araguas, L., Gonfiantini, R., 2013. Isotopic patterns in modern
global precipitation. Geophys. Monogr. Ser. 1–36.
Song X., Loucos K.E., Simonin K.A., Farquhar G.D., Barbour M.M., 2015a.
Measurements of transpiration isotopologues and leaf water to assess enrichment
models in cotton. New Phytologist 206, 637–646.
Song X, Simonin KA, Loucos KE, Barbour MM, 2015b. Modelling non-steady-state
isotope enrichment of leaf water in a gas-exchange cuvette environment. Plant, Cell
and Environment 38, 2618-2628.
Schefuβ, E., Kuhlmann, H., Mollenhauer, G., Prange, M., Pätzold, J., 2011. Forcing of
wet phases in Southeast Africa over the past 17,000 year. Nature 480, 22–29.
Schleinger WH, Jasechko S, 2014. Transpiration in the global water cycle. Agr. Forest
Meterol. 189-190, 115-117.
Sprenger M, Leistert H, Gimbel K, Weiler M, 2016. Illuminating hydrological
processes at the soil-vegetation-atmosphere interface with water stable isotopes. Rev.
Geophys. 54, 674-704.





Sprenger M, Tetzlaff D., Soulsby S., 2017. Soil water stable isotopes reveal evaporation
dynamics at the soil-plant-atmosphere interface of the critical zone. Hydrol. Earth Syst.
Sci. 21, 3839-3858.
Wang J, Fu B, Lu N, Zhang L, 2017. Seasonal variation in water uptake patterns of
three plant species based on stable isotopes in the semi-arid Loess Plateau. Science of
the Total Environment 609, 27-37.
Wang, J., Lu, N., Fu, B., 2019b. Inter-comparison of stable isotope mixing models for
determining plant water source partitioning. Sci. Total Environ. 666, 685–693.
Wu, H., Li, J., Li, X., He, B., Liu, J., Jiang, Z., Zhang, C., 2018. Contrasting response
of coexisting plant′s water-use patterns to experimental precipitation manipulation in
an alpine grassland community of Qinghai Lake watershed, China. PLoS One 13,
e0194242.
Wu H, Wu J, Sakiev K, Liu J, Li J, He B, Liu Y, Shen B, 2019. Spatial and temporal
variability of stable isotopes ($\delta^{18}$O and $\delta^2$H) in surface waters of arid, mountainous
Central Asia. Hydrological Processes 33, 1658-1669.
Yang, Y.G., Fu, B.J., 2017. Soil water migration in the unsaturated zone of semiarid
region in China from isotope evidence. Hydrol. Earth Syst. Sci. 21, 1–24.
Zhang P, Liu W, 2010. Effect of plant life form on relationship between $\delta$D values of
leaf wax n-alkanes and altitude along Mount Taibai, China. Organic Geochemistry 42,

614    100-107.

Zhao L, Wang L, Cernusak LA, Liu X, Xiao H, Zhou M, Zhang S, 2016. Significant
difference in hydrogen isotope composition between xylem and tissue water in *Populus*





*Euphratica*. Plant Cell Environ 39, 1848-1857.
Zhao, Y., Wang, Y., He, M., Tong, Y., Zhou, J., Guo, X., Liu, J., Zhang, X., 2020.
Transference of *Robinia pseudoacacia* water-use patterns from deep to shallow soil
layers during the transition period between the dry and rainy seasons in a waterlimited
region. For. Ecol. Manag. 457, 117727.
Zhang, H., Cheng, H., Cai, Y., Spötl, C., Sinha, A., Kathayat, G., Li, H., 2020. Effect of
precipitation seasonality on annual oxygen isotopic composition in the area of spring
persistent rain in southeastern China and its paleoclimatic implication. Climate of the
Past 16, 211-225.
Zhang, H., Zhang, X., Cai, Y., Sinha, A., Spötl, C., Baker, J., Kathayat, G., Liu, Z., Tian,
Y., Lu, J., 2021. A data-model comparison pinpoints Holocene spatiotemporal pattern
of East Asian summer monsoon. Quaternary Science Reviews 261, 106911.


**Figure captions**
Fig. 1 Sample sites (black dots) and vertical disrtibution of vegetation across the Mt. Taibai transect
(originating from Liu, 2021).
Fig. 2 Boxplots of precipitation, soil water, stem water, and leaf water for $\delta^{18}O$ values (a-d) and $\delta^2H$
values (e-h). Box plots show the median (red line), interquartile range (IQR) with the upper (75%)
and lower (25%) quartiles, lowest whisker still within 1.5 IQR of th elower quartile, and highest
whisker still within 1.5 IQR of the upper quartile; dots mark outliers.
Fig. 3 Dual isotope plots of precipitation, soil water, stem water, and leaf water in May (a), July (b),
and September (c).
Fig. 4 Relationships between altitude and $\delta^{18}O$ (a-c) and $\delta^2H$ values (d-f) from different water types
across months.
Fig. 5 Variation of monthly mean precipitation $\delta^{18}O$ (a) and $\delta^2H$ (b) values at Xi'an station from
Global Network of Isotopes in Precipitation (GNIP) and cluster mean of moisture transport routes
using HYSPLIT model in May (c), July (d) and September (e), 2020. Background in (c-e) is the
average precipitation (mm/day) and 850 hPa wind vectors (arrows, m/s) in May (c), July (d) and
September (e) in 1979-2016 AD based on the database of the Global Precipitation Climatology





Center (GPCC) (Becker et al., 2011) and the Modern-Era Retrospective analysis for Research and
Applications (Rienecker et al., 2011).
Fig. 6 Dual isotope plots for leaf water across month and altitude.
Fig. 7 Isotopic schematics of the flow diagram from precipitation to leaf water. Overview of the
processes through multiple isotopic fractionations associated with various hydroclimate and physio-
biological factors in terrestrial plants (Modified from Sachse et al., 2012 and Liu et al., 2016).
















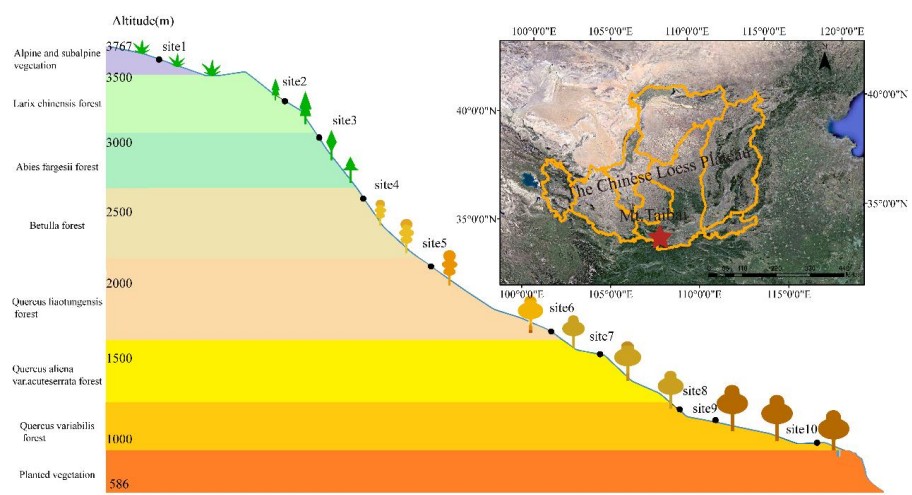


Figure-1






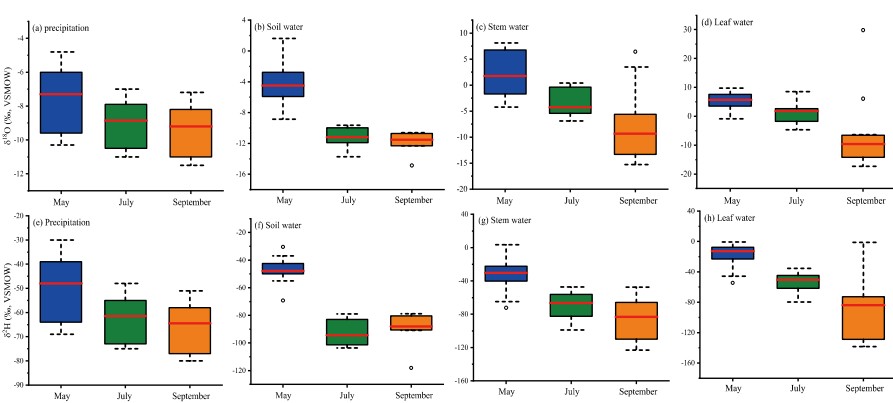

Figure-2

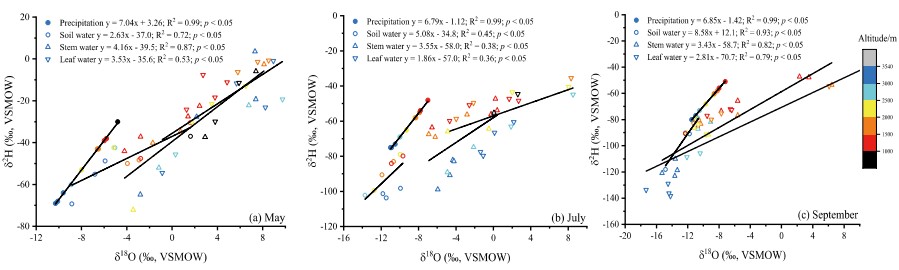

Figure-3

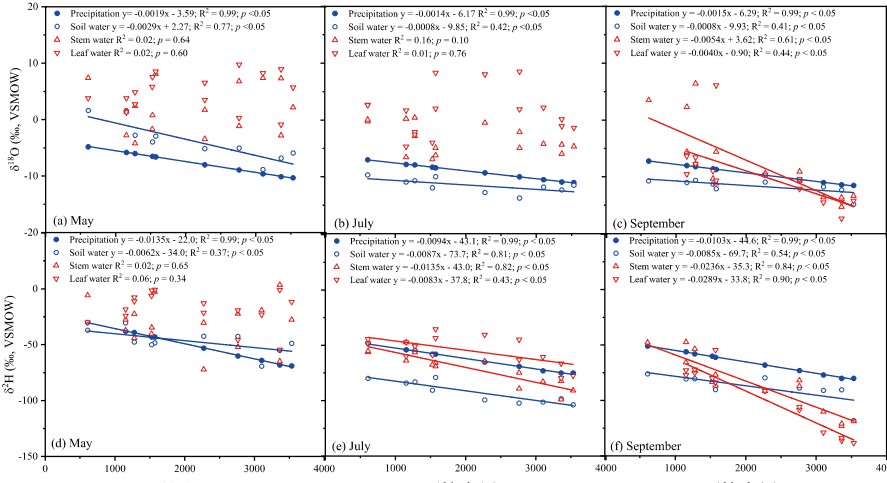

Figure-4





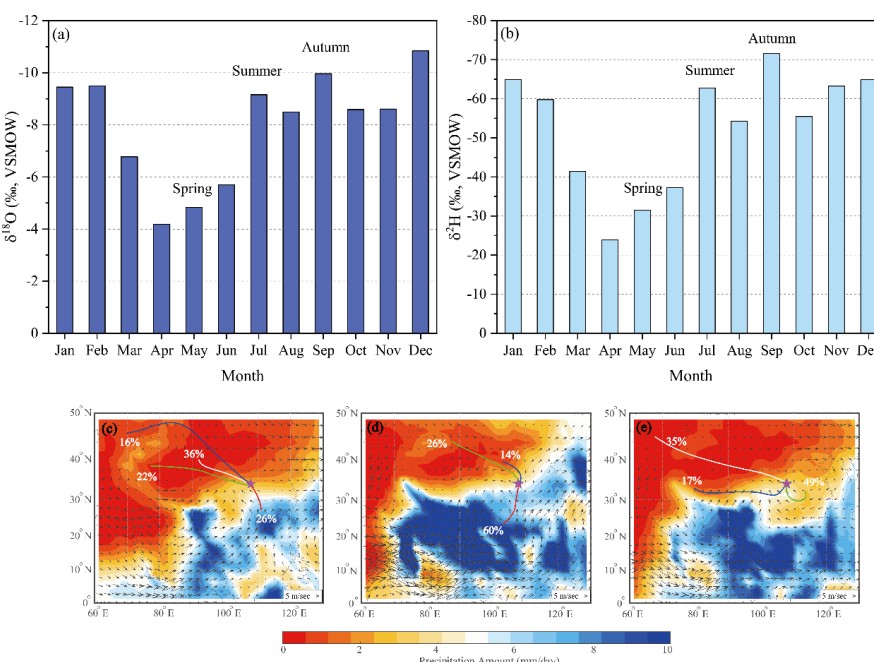

Figure-5

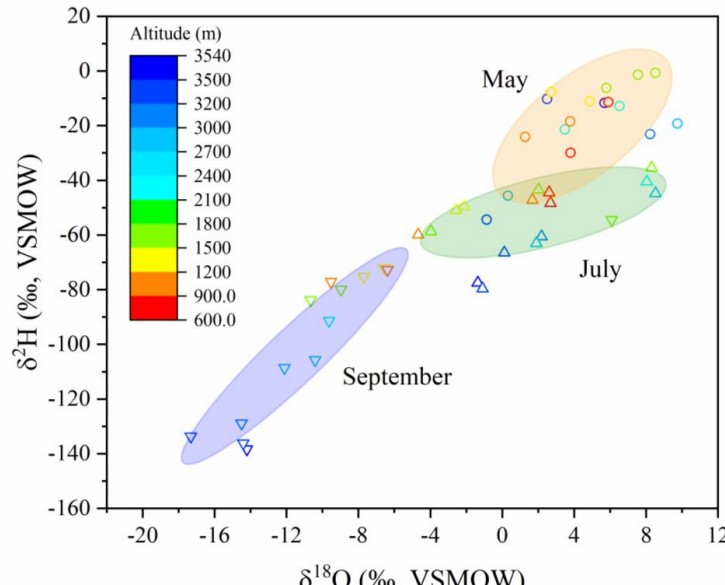

Figure-6





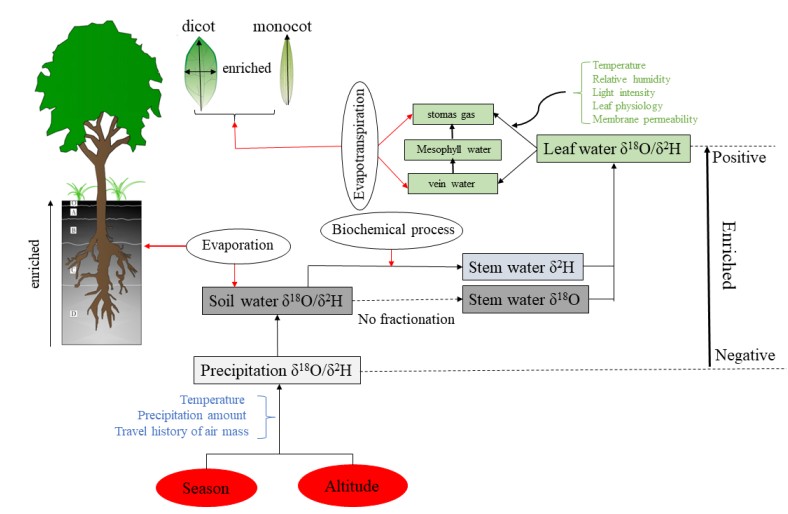


Figure-7
