# Peer review of "Controls of seasonality and altitude on generation of leaf water isotopes"

_Hydrology and Earth System Sciences, 2021_

## Author Comment (AC1)

Comment

*This manuscript tried to address how leaf water isotopes generate to be an isotopic line, which is important in ecohydrological cycle. The authors thought that both altitude and seasonality likely control to result in this line generation by repeatedly sampling soil water, stem water, and leaf water across altitudinal transect. The selection of topic, design of experiment, and conclusion were interesting, which was not yet reported so far. I like this manuscript and support the publication of this manuscript after addressing a minor revision below*

Response:

Thanks for the review's positive comments on our work. We will address all the comments point-by-point.

Comment

1. *A growing number of studies have, recently, reported that there was a large isotopic offset between root/xylem water to soil water, challenging the prior assumption that on isotopic fractionations during root water uptake occurred. Did your results support this view? If the isotopic offsets existed, how did they affect your results?*

Response:

Thank you for your good question. We added more results and discussion on isotopic offsets between stem water and soil water. The isotopic offsets have been reported in several studies recently, which challenge the prior assumption of no isotope fractionation during root water uptake by terrestrial plants (Lines 228-231; 396-410).

Comment

2. *Your results showed an isotopic consistency from precipitation, soil, stem, and leaf waters, so you think that leaf water isotopic line were generated by the first-order control of source water. I am curious about whether other potential water sources (e.g., fog, dew) affect the results?*

Response:

Thank you. It is really a good point.

We will do more investigations for other potential water sources, and soil water mobile and immobile water. The systematic investigation for waters will help us to better understand ecohydrological processes in the forestland and grassland.

Comment

*Some minor revisions as below*

*Line 24, bridges →bridge;*

Response:

Thank you. We have revised it.

Comment

*Line 30: deplete "the" between "in" and "δ18O";*

Response:

Thank you. We have revised it.

Comment

*Line 54: deplete "the" before "water balance";*

Response:

Thank you. We have revised it.

Comment

*Line 55-56: The statement on enriched isotopic compositions of leaf water resulted from evapotranspiration, which is doubtful. In fact, this isotopic enrichment is mainly caused by evaporation whereas the evapotranspiration is generally considered no isotopic fractionation. Please check this statement:*

Response:

Thank you. We have revised it.

Comment

*Line 67: Change "first" to "firstly";*

Response:

Thank you. We have revised it.

Comment

*Line 71, controls →controlling;*

Response:

Thank you. "Controls" is a noun in the location.

Comment

*Line 83-86, the isotopic fractionation by roots should be removed because this study did not refer this relevant data and experimental design;*

Response:

Thank you. We have revised it.

Comment

*Line 88: Change "insight" to "insights";*

Response:

Thank you. We have revised it.

Comment

*Line 140, This part should be listed the detailed information on the plots. The experimental design is very vital to determine the monitoring data of this study. Plant and soil properties should be explained in the section. According to this, the study can select which plant and soil samples can be collected. Hence, line 143, one or two plant species refer to which plants? This plant represents the typical? Shrub? Trees?*

Response:

Thank you. We have

Comment

*Line 146 remove the minimal damage;*

Response:

Thank you. We have revised it.

Comment

*Line 150, why do you collect samples from surface soil layers (< 10cm)? Other deep soil layers? Plant use water sourced from not only surface water but deep soil layers;*

Response:

Thank you.

Comment

*Line 231: Missing something, rephrase the sentence;*

Response:

Thank you. We have revised it.

Comment

*Line 232-234, this statement moves to the discussion section.*

Response:

Thank you. We have moved the statements to the discussion section.

---

## Author Comment (AC2)

Comment

*Liu et al present a manuscript titled "controls of seasonality and altitude on generation of leaf water isotopes". Leaf water isotopes have wide application in the hydrology and ecology. It is quite interesting to know the effects of seasonality and altitude on isotope variations of leaf water. This manuscript reported the isotopic variations of leaf water, stem water, soil water and rain water with the season and elevation. They concluded seasonality and altitude exert the influence on leaf water isotopes through precipitation as source water by comparing the seasonal and altitude dynamic of isotope compositions of those four types of water. However, the precipitation in figures 2-4 did not show significant consistent seasonal or altitude dynamics of the δ18O and δ2H values with leaf water. These data are also not able to support the figure 7. In my opinion, compared to effects of seasonality and altitude on precipitation, they are more likely to influence the environmental factors such as humidity and temperature, and then affect the generation of leaf water isotopes. I don't understand why the authors ignored to discuss other factors but only source water.*

Response:

Thank you for your approvedness and good suggestions. We have revised presentation, added more discussion and adjusted the related figures. See below:

As the abovementioned Craig-Gordon model was used to explain the variation in $\delta^{18}O$ and $\delta^2H$ values of leaf water, besides $\delta^{18}O$ and $\delta^2H$ values of source water (e.g., precipitation, soil water), the isotopic composition of water vapor, the equilibrium fractionation ($\varepsilon^+$) associated with temperature (Bottinga and Craig, 1969) and kinetic fractionation ($\varepsilon_k$) associated with stomatal and boundary resistances to water vapour (Farquhar et al., 2007) and climatic factors (temperature, relative humidity) also potentially affect leaf water $\delta^{18}O$ and $\delta^2H$ values. Additionally, both seasonality and altitude play an effect on precipitation $\delta^{18}O$ and $\delta^2H$ values through affecting climatic factors (temperature and relative humidity). The soil evaporation, leaf transpiration and biochemical processes associated with multiple climatic and physiological factors exert a secondary effect on leaf water $\delta^{18}O$ and $\delta^2H$ values. The precisely mathematic equation that control leaf water $\delta^{18}O$ and $\delta^2H$ values under non-steady-state conditions and the the *Péclet* effect remain to be further studied (Song et al., 2015b; Cernusak et al., 2016; Barbour et al., 2017).

Comment

*Besides, some unexpectable data were not explained such as much enriched stem water relative to soil water. Moreover, some key information such as the sampling time, plant names, how many species totally are presented in the figures, each symbol represents one plant or one species?*

Response:

Thank you. We have supplemented the explanation for the offset between stem water and soil water (Lines 193-196; 394-408). Also, we added more information for sampling and species (Lines: 142-152; details in Supplementary Table S1). See below:

Our results showed significant isotopic offsets between stem water and soil water, not just in $\delta^2H$ stem-soil water offset ($\Delta^2H$; Fig. 3b), but also in $\delta^{18}O$ stem-soil water offset ($\Delta^{18}O$; Fig. 3a). The significant isotopic offsets ($\Delta^{18}O$ and $\Delta^2H$) demonstrates that stem-soil isotopic fractionation is not restricted to halophytes (Lin and Sternberg, 1993; Ellsworth and Williams, 2007; Redelstein et al., 2018) or xerophytes (Ellsworth and Williams, 2007; Zhao et al., 2016) but is likely more common and can also occur in temperate alpine forests (Fig. 3). The results were also observed in recent studies over a range

of climates and biomes (Zhao et al., 2016; Wang et al., 2017; Barbeta et al., 2019; Poca et al., 2019; Liu et al., 2021a). The long-standing principle of no isotopic fractionation during root water up likely requires reconsideration (Barbeta et al., 2020). The isotopic offsets are considered to likely originate from methodological flows, e.g., laser-based instruments (Martín-Gómez et al., 2015; Millar et al., 2018) or cryogenic vaccum extraction artifacts (Orlowski et al., 2018; Chen et al., 2020), arbuscular mycorrhizal associations (Poca et al., 2019), soil water pools exchanges under dry and wet conditions (Barbeta et al., 2020).

Comment

*Basically, this manuscript reported fairly important data and is addressing a topic which would interest large-scale ecosystem researchers. I suggest the authors reorganise this manuscript, confine their conclusions to their data. My detailed comments are*

Response:

Thank you for your approvedness on our work. We have revised the discussion and conclusion.

Comment

*Line 1, change "Controls" to "effect"*

Response:

Thank you. We have revised it.

Comment

*Line 30-32, it will be clearer if change to" consistent seasonal dynamics of the δ18O and*
*δ2H values in precipitation, soil water, stem water, and leaf water"*

Response:

Thank you. We have revised it.

Comment

*Line 35, change to "which result in"*

Response:

Thank you. We have revised it.

Comment

*Line 39, please rewrite the summary: first, it is difficult to understand those sentences; second, the linearity of dual-isotope plot is basically results of the equilibrium and kinetic fractionation factors*

Response:

Thank you. We have revised the summary.

Comment

*Line 40, rewrite, may be "Why can dual-isotope plot of spatiotemporal leaf water be linear"*

Response:

Thank you. We have revised it.

Comment

*Line 41-42, confused, the LMWL is also an isotopic water line*

Response:

Thank you. We have revised it.

Comment

*Line 44-45, change to "effects of seasonality and altitude on source water*

Response:

Thank you. We have revised it.

Comment

*Line 73, add "More specifically" before "Soil water isotopes"*

Response:

Thank you. We have added it.

Comment

*Line 143, what plants? trees or grasses?*

Response:

Thank you. We have listed the information for samples in Supplementary Table S1

Comment

*Line 145-146, how many leaves for one plant? What time were those leaves collected?*
*Diurnal variations of leaf water isotopes could be over those caused by the season or*
*elevation.*

Response:

Thank you. We have added more information for plant sampling.

Yes, I agree with the diurnal variation of leaf water isotopes. In order to reduce or exclude the effect of diurnal variation of leaf water isotopes, we collected plant leaves between 12 pm to 15 pm for three sample campaigns.

Comment

*Line 165-167, the laser isotope analyzer is quite sensitive to the organic matter in the soil.*
*don't know whether you did the calibration or not?*

Response:

Yes, it is a good question. In this study, we measured soil water isotopes using the laser isotope analyzer (Picarro L2130-I isotope water analyzer), and we measured root and leaf water isotopes using an isotope ratio mass spectrometer coupled with a high-temperature conversion elemental analyzer (HT2000 EA-IRMS, Delta V Advantage; Thermo Fisher Scientific).

Comment

*Line 180-181, how the authors could convince us this online system can work in this study area.*
*Here the spatiotemporal resolutions should be stated.*

Response:

Thank you. The Online Isotope in Precipitation Calculator (OIPC) model is relatively reliable for determining $\delta^{18}O$ and $\delta^2H$ values of precipitation at low and middle latitudes based on the Global Network for Isotopes in Precipitation (GNIP) database (Bowen and Revenaugh, 2003; Sachse et al., 2004;

Tipple and Pagani, 2013; Daniels et al., 2017).

**References**

Bowen, G.J., Revenaugh, J., 2003. Interpolating the isotopic composition of modern meteoric precipitation. Water Resour. Res. 39, 1299.

Sachse, D., Radke, J., Gleixner, G., 2004. Hydrogen isotope ratios of recent lacustrine sedimentary n-alkanes record modern climate variability. Geochim. Cosmochim. Acta, 68, 4877–4889.

Tipple, B.J., Pagani, M., 2013. Environmental control on eastern broadleaf forest species' leaf wax distributions and D/H ratios. Geochim. Cosmochim. Acta 111, 64–77.

Daniels, W.C., Russell, J.M., Giblin, A.E., Welker, J.M., Klein, E.S., Huang, Y., 2017. Hydrogen isotope fractionation in leaf waxes in the Alaskan Arctic tundra. Geochim. Cosmochim. Acta 213, 216–236.

Comment

*Line 237, why are O isotope values of stem water quite higher than the soil water and rain water? The authors should explain this in Fig 4.*

Response:

Thank you. We have added more discussion about stem-soil isotopic offsets in Section 4.3.

Comment

*Line 246, the fig 2 did not show the statistical significance. It seems the precipitation has no seasonal variation. Also, are those boxplots the summaries of whole-elevation samples?*

Response:

Thank you. We have revised the presentation as you suggested. In Fig. 2, seasonal variation in precipitation isotopes are likely not significant across months, but in Fig. 5, we can see significant seasonal variation. Also, we also test the significance across month by a one-way ANOVA test.

Yes, the boxplots showed the all samples covering different elevations.

Comment

*Line 349-351, non-steady state model of leaf water isotope states the isotopes of source water, isotopes of ambient vapor, humidity and temperature, and transpiration and leaf traits determine isotope values of leaf water. Theoretically, source water is related to precipitation, but in this study either the seasonality or altitude of the precipitation isotopes are hardly related to leaf water or stem water (fig1&3). Also, the dule-isotope plot of did not show significant seasonal dynamics as that of leaf water (fig2). Therefore, those data could not support the claim (fig6). Compared to effects of seasonality and altitude on precipitation, they are more likely to influence the environmental factors such as humidity and temperature. I don't understand why the authors ignored to discuss other factors but only source water.*

Response:

Thank you. We have revised the discussion (Lines 378-393).

Comment

*Lin 355-364 those are the general significance, not this paper's Insights and implications.*
*Delete or put them in introduction*

Response:

Thank you. We have delete the paragraph, but we added more discussion on the isotopic offsets, which are being observed in some studies (Zhao et al., 2016; Wang et al., 2017; Barbeta et al., 2019; Barbeta et al., 2020; Poca et al., 2019; Liu et al., 2021a).

**References**

Barbeta A, Jones SP, Clavé L, Gimeno TE, Fréjaville B, Wohl S, Ogée J, 2019. Unexplained hydrogen isotope offsets complicate the identification and quantification of tree water sources in a riparian forest. Hydrol Earth Syst Sci 23, 2129-2146.

Barbeta A, Gimeno TE, Clavé L, Fréjaville B, Jones SP, Delvigne C, Wingate L, Ogée J, 2020. An explanation for the isotopic offset between soil and stem water in a temperate tree species. New Phytologist 227, 766-779.

Wang J, Fu B, Lu N, Zhang L, 2017. Seasonal variation in water uptake patterns of three plant species based on stable isotopes in the semi-arid Loess Plateau. Science of the Total Environment 609, 27-37.

Zhao L, Wang L, Cernusak LA, Liu X, Xiao H, Zhou M, Zhang S, 2016. Significant difference in hydrogen isotope composition between xylem and tissue water in Populus Euphratica. Plant Cell Environ 39, 1848-1857.

Poca M, Coomans O, Urcelay C, Zeballos SR, Bodé S, Boecks P, 2019. Isotope fractionation during root water uptake by Acacia caven is enhanced by arbuscular mycorrhizas. Plant and Soil, 441, 485–497.

Liu J, Wu H, Cheng Y, Jin Z, Hu J, 2021a. Stable isotope analysis of soil and plant water in a pair of natural grassland and understory of planted forestland on the Chinese Loess Plateau. Agricultural Water Management 249, 106800.

Comment

*Line 638-639, does each symbol represents one plant or one species?*

Response:

Thank you. Each symbol represents an individual plant (The data in seeing Supplementary Table S1).

Comment

*Line 640-641, same as above.*

Response:

Thank you. Each symbol represents an individual plant (The data in seeing Supplementary Table S1).

Comment

*Line 649, same question as above. What is the difference of this figure from the figure 3?*

Response:

Thank you. Each symbol represents an individual plant (The data in seeing Supplementary Table S1). The Fig. 4 is to show the isotopic relationships among different waters, but in Fig. 7 it showed the effects of seasonality and altitude on leaf water dual-isotope linearity, which is also the topic of the article.

Comment

*Line 650-652, I don't think those data of this study support this figure.*

Response:

Thank you. We have revised the figure.